# Rice Heat Stress Response: Physiological Changes and Molecular Regulatory Network Research Progress

**DOI:** 10.3390/plants14162573

**Published:** 2025-08-19

**Authors:** Weiwei Ma, Xiaole Wang, Chuanwei Gu, Zhengfei Lu, Rongrong Ma, Xiaoyan Wang, Yongfa Lu, Kefeng Cai, Zhiming Tang, Zhuoqi Zhou, Zhixin Chen, Huacheng Zhou, Xiuhao Bao

**Affiliations:** 1Institute of Crop Sciences, Ningbo Academy of Agricultural Sciences, Ningbo 315000, China; jonathan_@163.com (W.M.); linda2086415@163.com (X.W.); lzf202508@163.com (Z.L.); nbseed001@sina.com (R.M.); wxy900@163.com (X.W.); luyongfa2013@163.com (Y.L.); 13806672665@163.com (K.C.); tzm2004@163.com (Z.T.); 2State Key Laboratory for Crop Genetics and Germplasm Enhancement, Jiangsu Plant Gene Engineering Research Center, Nanjing Agricultural University, Nanjing 210095, China; 15301584653@163.com; 3Ningbo Seed Co., Ltd., Ningbo 315000, China; oshipipi1@163.com (Z.Z.); nbseedczx@163.com (Z.C.)

**Keywords:** rice, heat stress, physiological function, molecular regulatory network, heat tolerance

## Abstract

Global climate change has markedly increased the frequency of heat stress events in rice, severely threatening both yield and grain quality and posing a substantial challenge to global food security. Understanding the molecular mechanisms underlying heat tolerance in rice is therefore essential to facilitate the breeding of thermotolerant cultivars. This review provides a comprehensive overview of the effects of heat stress on rice agronomic traits across various developmental stages. We summarize key physiological and metabolic alterations induced by high temperatures and discuss recent advances in unraveling the molecular regulatory networks involved in heat stress responses. By integrating findings from gene cloning, functional genomics, and advanced breeding strategies, this review outlines practical approaches for improving rice heat tolerance and identifies critical knowledge gaps that warrant further investigation.

## 1. Introduction

Rice (*Oryza sativa* L.) is a principal staple crop, serving as the primary food source for over half of the global population. However, in the context of global warming, the increasing frequency and severity of extreme high-temperature events have led to substantial reductions in rice productivity. Heat stress (HS) adversely affects rice yield, grain quality, and harvest index, posing a critical threat to food security [1,2]. Studies have shown that for every 1 °C increase in average temperature during the rice growing season, rice yield is expected to decrease by approximately 6.2%. Projected climate change could reduce total brown rice production by 7.1–8.0% and milled rice yield by 9.0–13.8%, resulting in an estimated 8.1–11.0% decrease in economic returns from rice processing [2,3]. High temperatures impair rice growth by shortening the growth period, reducing photosynthetic efficiency, increasing respiratory rates, and accelerating transpiration. These physiological changes can lead to the accumulation of reactive oxygen species (ROS), a reduction in biomass, and impaired capacity for organic compound synthesis [4]. The reproductive stage of rice is particularly vulnerable to HS. Exposure to HS during this period can reduce pollen viability, cause gametophyte sterility, and result in severe fertilization failure, ultimately leading to yield losses. When high temperatures occur during grain filling, the source–sink energy transfer becomes disrupted. This impairs endosperm development and reduces the accumulation of essential nutrients, including starch, proteins, and trace elements. As a result, grain quality is significantly diminished [4,5]. Heat stress has also been shown to disrupt hybrid seed production by interfering with flowering synchronization, reducing pollen fertility, and increasing embryo sac abnormalities. Such reproductive failures pose significant challenges for both rice breeding programs and hybrid seed production in the context of climate change. A clear understanding of the physiological and molecular mechanisms of rice responses to HS is crucial. Such knowledge will help identify heat-tolerance genes and accelerate the development of resilient cultivars, safeguarding global food security under climate change.

To ensure a comprehensive and high-quality synthesis, we conducted a structured literature search using major scientific databases including Web of Science, PubMed, and Google Scholar. Publications from January 2000 to May 2025 were considered, with keywords such as “rice,” “heat stress,” “thermotolerance,” “QTL,” “gene expression,” “physiology,” and “molecular regulation” used individually and in combination. Only peer-reviewed articles published in English were included. Priority was given to studies providing clear experimental evidence relevant to physiological responses, molecular mechanisms, and breeding strategies for heat tolerance. Review articles were consulted to provide context, but the primary emphasis was placed on original research.

## 2. Impact of Heat Stress on the Agronomic Performance of Rice

High temperatures adversely affect rice at all developmental stages, with the vegetative and reproductive phases being particularly vulnerable (Table 1). Elevated temperatures disrupt processes ranging from seed germination and seedling establishment to tillering and reproductive development. Critical reproductive impairments, such as abnormal anther dehiscence, reduced pollen viability, and defective embryo sac formation, collectively compromise yield and grain quality (Figure 1). Therefore, a comprehensive investigation of HS effects on agronomic traits, along with detailed knowledge of stage-specific responses, is essential for breeding heat-tolerant cultivars and ensuring stable rice production.

High temperature stress significantly affects rice development throughout its life cycle, particularly during the vegetative and reproductive growth phases. At the seedling and tillering stages, HS inhibits germination, reduces seedling vigor, impairs root and shoot development, and limits tiller formation. In the reproductive phase, which includes panicle initiation, booting, flowering, and grain filling, elevated temperatures can disrupt meiosis, cause pollen abortion, reduce spikelet fertility, lead to abnormal embryo sac development, and impair grain filling. These physiological disruptions ultimately result in lower yields and reduced grain quality. A stage-specific understanding of heat-induced agronomic impacts is critical for improving heat tolerance in rice.

### 2.1. Impact of Heat Stress on the Vegetative Growth Stage of Rice

Under prolonged high-temperature conditions, seed germination rate and vigor are reduced, and the development of both the plumule and radicle is inhibited. In addition, photosynthetic efficiency in rice seedlings declines, accompanied by reduced chlorophyll content and a significant slowdown in growth rate. These changes also impair the seedlings’ ability to absorb water and nutrients, often resulting in symptoms such as leaf chlorosis, leaf tip necrosis, and wilting. The critical temperature threshold for heat response in rice seedlings is 35 °C; temperatures above this level may lead to seedling mortality. Furthermore, high temperatures weaken the seedlings disease resistance, increasing the incidence of diseases and the seedling mortality rate [6,7,8,9]. The tillering stage is a critical period for determining the number of panicles, which is a major component of rice yield. Studies have shown that within the temperature range of 16 °C to 35 °C, higher temperatures promote faster tiller development. However, when the temperature exceeds the optimal range, tillering may cease or the maximum number of tillers may be reduced, ultimately leading to a decrease in yield. In addition, excessive temperatures negatively affect dry matter accumulation, leaf area index, and plant height during the tillering stage, thereby inhibiting overall plant growth [10].

### 2.2. Effects During the Reproductive Growth Stage

The panicle initiation stage is a critical phase of reproductive development in rice. HS during this period can severely disrupt the development of pollen mother cells, particularly during the process of meiosis. In addition, high temperatures may induce abnormalities in microspore development and, in some cases, lead to developmental arrest, ultimately resulting in male sterility. These damages have irreversible effects on the reproductive capacity of rice [11,12]. High temperatures impact rice yield by disrupting physiological processes. Elevated daytime temperatures reduce photosynthesis, while high nighttime temperatures increase nutrient consumption, limiting nutrient supply to developing spikelets. During the nutrient-demanding panicle initiation stage, HS reduces nutrient accumulation, decreasing spikelet number, grains per panicle, and seed-setting rate [13].

In addition, high temperatures significantly affect grain morphology by reducing the length, width, and thickness of the rice hulls [14]. High temperatures also reduce the accumulation of dry matter in the panicle and limit the translocation of pre-anthesis assimilates from vegetative organs to the developing panicle. Instead, more dry matter tends to be retained in the vegetative tissues, thereby weakening nutrient supply to the grains. As a result, final grain weight, spikelet number per panicle, and seed-setting rate are significantly reduced [15]. Collectively, HS during panicle initiation impairs rice reproductive capacity and yield formation by disrupting reproductive organ development, compromising photosynthesis and nutrient supply, altering grain morphology, and limiting assimilate translocation.

The heading-up stage is a critical phase in rice growth during which the panicle fully develops and emerges from the flag leaf sheath. Heat stress during this period significantly inhibits the normal elongation of sheath cells, leading to abnormal heading or even complete blockage of the heading process. This directly affects subsequent flowering and grain filling [16]. As the panicle emerges, flowering (anthesis) typically occurs on the same day or within the following few days. The anthesis period is one of the most critical stages in rice reproductive growth and is also the most heat-sensitive. Heat stress during this period primarily affects the normal functioning of pollen grains and anthers, leading to pollination and fertilization barriers. On one hand, high temperatures reduce pollen viability and weaken its expansion ability, making it difficult for the anthers to dehisce properly. On the other hand, high temperatures may also cause the closure of the anther cell layers, hindering pollen grain release and further reducing pollination efficiency [17]. Heat stress adversely affects the maternal organ tissues of rice, primarily manifesting in abnormal development of the pistil. High temperatures cause issues during the differentiation of megaspore mother cells in the IR64 variety, with some cells failing to differentiate correctly. In some cases, all four megaspores degenerate after meiosis, leading to abnormal development of the embryo sac [18]. In addition, HS induces proliferative abnormalities in pistil tissues, including the formation of multiple stigmas and/or ovaries [19], all of which may negatively affect fertilization and grain development in rice.

During the grain-filling stage in rice, from flowering to grain maturity, HS reduces yield and grain quality by shortening the filling duration and suppressing photosynthate synthesis and translocation, leading to lower organic substance accumulation and yield. High temperatures inhibit key enzyme activity and gene expression, reducing sucrose-to-starch conversion and starch accumulation, altering the amylose-to-amylopectin ratio, and impairing eating and processing quality. Early-stage HS significantly reduces amylose accumulation, while late-stage stress may slightly enhance it [20]. High temperatures during grain filling shorten the filling period, causing loose and uneven starch granule arrangement, increasing the broken rice rate, and reducing brown and milled rice rates. HS impairs appearance quality by decreasing grain plumpness, increasing chalkiness, and reducing translucency, lowering market competitiveness and economic value. Additionally, it reduces starch crystallinity, leading to uneven granule surfaces and indentations, which decrease gel consistency, soften texture, and diminish eating and cooking quality [21].

## 3. Physiological and Biochemical Alterations Induced by Heat Stress in Rice

Under HS, the physiological changes in rice are primarily manifested as membrane damage, ROS accumulation, impaired photosynthesis, metabolic abnormalities, and hormone imbalances (Figure 2). These changes interact with each other, affecting the normal growth and development of rice.

Major physiological disruptions in rice under HS include membrane damage, ROS accumulation, impaired photosynthesis, and altered carbohydrate and hormone metabolism. ROS accumulation and membrane injury compromise cellular integrity and trigger PCD. Photosynthetic capacity declines due to thylakoid disintegration, chlorophyll loss, and Rubisco/RCA inactivation, reducing carbon fixation. Heat stress also disrupts the TCA cycle and starch biosynthesis, leading to lower ATP production, increased grain chalkiness, and poor quality. Hormonal imbalances, such as excess ethylene and ABA, exacerbate oxidative stress, promote stomatal closure, and further limit photosynthesis, ultimately reducing growth, yield, and reproductive success.

### 3.1. Membrane Damage and ROS Accumulation

The plasma membrane (PM) plays a crucial role in plant responses to HS, particularly in maintaining membrane stability and fluidity under elevated temperatures. The PM serves as the primary HS sensor, mediating rapid temperature perception and downstream signal transduction. High temperatures alter membrane fluidity and stability, thereby activating membrane-associated sensors such as calcium ion channels. This activation leads to a rapid influx of Ca^2+^, triggering downstream signaling cascades. This process is vital for plant cells to perceive temperature fluctuations and initiate adaptive mechanisms. Under HS, changes in membrane lipid saturation and fatty acid composition further affect membrane fluidity and integrity, thereby influencing the membrane’s functional stability [22,23].

In addition, HS stimulates the excessive production of ROS within rice cells. Excess ROS cause widespread and severe damage to cellular components. One major consequence is lipid peroxidation of the cell membrane, in which ROS oxidize membrane lipids and exacerbate membrane damage. This oxidative stress leads to increased levels of malondialdehyde (MDA), a key marker of lipid peroxidation, and impairs the normal function of proteins and nucleic acids. In severe cases, ROS accumulation can trigger programmed cell death (PCD), further compromising plant cell viability under HS [24,25]. Furthermore, HS also compromises the activity of antioxidant enzymes, particularly superoxide dismutase (SOD) and catalase (CAT), which play vital roles in scavenging ROS and maintaining cellular redox balance. The decline in these enzymatic activities under high temperature conditions further exacerbates oxidative damage, contributing to increased cellular injury and dysfunction [26,27].

### 3.2. Photosynthetic Impairment

High temperature stress significantly impairs the photosynthetic performance of rice through multiple interconnected mechanisms [28]. Primarily, it disrupts the structural integrity of photosystem II (PSII), particularly affecting the oxygen-evolving complex (OEC) and D1 protein, leading to decreased electron transport efficiency and ROS accumulation, which exacerbates PSII photoinhibition. Moreover, high temperatures suppress linear electron flow (LEF), prompting the activation of cyclic electron flow (CEF) as a compensatory mechanism to enhance thermal dissipation and ATP synthesis, thereby partially protecting the photosynthetic apparatus [29]. HS can disrupt the permeability of thylakoid membranes and even cause disintegration of thylakoid granules, leading to a reduction in chlorophyll content, disturbances in photochemical reactions, and a significant decline in the photosynthetic rate [30,31]. Furthermore, HS inhibits the activity of ribulose-1,5-bisphosphate carboxylase/oxygenase (Rubisco) and its activase (RCA), reducing carbon fixation efficiency and promoting photorespiration, which increases energy consumption and lowers net photosynthetic output. The decline in chlorophyll content and disruption of thylakoid membrane structure further weaken photosynthetic efficiency [32].

### 3.3. Energy Metabolism Imbalance

High temperature exerts a multifaceted impact on the tricarboxylic acid (TCA) cycle in rice by disrupting substrate availability, altering metabolic flux, inhibiting key enzymatic activities, increasing ROS production, and disturbing cellular energy homeostasis. A major upstream factor contributing to these effects is the impairment of photosynthesis. At approximately 38 °C, photosynthetic rates in rice decline by 30–50% [33,34,35], primarily due to damage to chloroplast membranes and thermal inactivation of essential Calvin cycle enzymes such as Rubisco and sedoheptulose-1,7-bisphosphatase. This inhibition of photosynthesis reduces carbohydrate production in source tissues, which consequently limits the supply of assimilates to sink organs. As a result, the availability of glycolytic substrates, particularly pyruvate and acetyl-CoA, is significantly reduced. Furthermore, high temperature suppresses the activity of glycolytic enzymes, with phosphofructokinase showing an approximately 25% reduction in activity under HS [36]. Fatty acid β-oxidation is also compromised [37], further diminishing acetyl-CoA production and weakening the input into the TCA cycle. Within the mitochondria, HS alters the metabolic flux through the TCA cycle. Intermediates such as citrate and isocitrate tend to accumulate, which is primarily attributed to the thermal inactivation of rate-limiting enzymes including α-ketoglutarate dehydrogenase and succinate dehydrogenase. These enzymes exhibit significant activity loss at elevated temperatures, such as 40 °C [38]. To counteract this damage, antioxidant systems are activated; however, these protective responses consume considerable energy, further exacerbating cellular energy deficiency [25,35]. Together, these factors—including inhibited photosynthetic carbon assimilation, reduced respiratory substrates, impaired TCA-cycle enzymes, and the high energy cost of ROS scavenging—greatly weaken mitochondrial energy production. These combined effects ultimately lead to impaired ATP synthesis, redox imbalance, and decreased growth, biomass accumulation, and yield in rice under HS. High temperature stress significantly affects rice seed development, particularly in terms of carbohydrate and starch metabolism. Studies have shown that during the development of rice grains after fertilization, several important sucrose hydrolyzing enzymes and transferases, such as Ugp1 (UDP-Glucose Pyrophosphorylase 1), SSIIa (Starch Synthase IIa), and OsTPP1 (Trehalose-6-Phosphate Phosphatase 1), are significantly inhibited under high temperatures. This leads to a reduction in sucrose and starch content in the affected grains [38,39]. Notably, high temperatures further decrease sucrose and starch content in rice seeds by inhibiting genes related to sucrose and starch synthesis. For example, key enzyme genes involved in starch synthesis, such as *cyPPDK* (*Cytosolic pyruvate*, *orthophosphate dikinase*), *GBSSI* (*Granule Bound Starch Synthase I*), and *BEIIb* (*branching enzyme IIb*), exhibit significant suppression under high-temperature conditions, limiting starch synthesis and ultimately impacting rice yield and quality [40,41]. Meanwhile, elevated temperatures also induce the expression of enzymes related to starch consumption and degradation, such as Amy1A (Alpha-Amylase 1A), Amy3D, Amy3E, SSSIIb, SSSIIc, SSSIIIb, and SSSIVa, leading to increased starch consumption and further affecting starch quality [42,43]. At the same time, HS leads to the accumulation of ROS, which not only inhibit carbohydrate synthesis but also compete with limited sugars to remove these toxic substances. This further disrupts the allocation of assimilates, reducing the amount of carbohydrates available for transport [44,45]. These changes have a negative impact on rice carbon metabolism, severely impairing seed energy supply and metabolic efficiency, and ultimately leading to the appearance of chalkiness in rice seeds.

### 3.4. Hormonal Imbalance

High temperature stress significantly affects the hormonal regulatory network in rice, involving multiple signaling pathways and physiological processes, thereby exerting profound impacts on rice growth, development, yield, and quality. It induces a notable imbalance in key phytohormones such as ethylene (ETH), abscisic acid (ABA), indole-3-acetic acid (IAA), and cytokinins (CTK), disrupting the normal physiological equilibrium [46].

Under HS, ethylene plays a dualistic role in modulating plant physiological responses. Studies have demonstrated that elevated temperatures significantly upregulate the expression of key ethylene biosynthesis genes, including 1-aminocyclopropane-1-carboxylic acid synthase (ACS) and ACC oxidase (ACO), thereby promoting excessive ethylene accumulation in plant tissues [47]. This hyperaccumulation of ethylene can intensify oxidative stress by enhancing the generation of ROS, which disrupts cellular redox homeostasis and accelerates membrane lipid peroxidation. In rice, such ethylene-induced oxidative stress contributes to compromised photosynthetic efficiency, accelerated leaf senescence, and inhibited root elongation, ultimately impairing overall plant growth and yield potential [48,49]. Despite its deleterious effects under chronic stress, ethylene may also activate certain defense pathways when tightly regulated, suggesting a complex and context-dependent regulatory mechanism. Thus, understanding the spatiotemporal dynamics of ethylene signaling and its crosstalk with antioxidant defenses and other hormonal pathways is crucial for elucidating rice heat tolerance mechanisms and for developing ethylene-modulating strategies to enhance crop resilience under climate-induced HS [47,50].

Heat stress significantly alters ABA dynamics in rice, with profound implications for plant physiology and stress tolerance. Under HS, ABA accumulation is rapidly induced, initially enhancing heat tolerance by promoting stomatal closure to minimize transpirational water loss and upregulating antioxidant defenses and heat shock proteins (HSPs) to mitigate oxidative damage [51]. For instance, moderate ABA levels bolster heat tolerance by increasing H_2_O_2_-mediated antioxidant capacity and enhancing activities of ascorbate peroxidase 1 and multiprotein bridging factor 1 [52]. Additionally, ABA facilitates sucrose transport and metabolism in rice spikelets, maintaining energy homeostasis by preventing ATP depletion, which is critical for reproductive success under HS [53]. However, prolonged or excessive ABA accumulation, particularly at high concentrations, disrupts metabolic balance, exacerbating oxidative stress through ROS overproduction and impairing photosynthetic efficiency by reducing net photosynthetic rates [54]. In rice cultivars with rolled leaves, elevated ABA levels further increase leaf temperatures due to reduced transpiration, intensifying respiratory energy demands and leading to severe metabolic imbalances [55]. Moreover, ABA-deficient mutants exhibit greater heat-induced damage compared to wild-type plants, underscoring ABA’s critical role, yet excessive ABA in sensitive cultivars or under prolonged HS weakens stress resilience by disrupting carbohydrate metabolism and energy allocation [56]. These contrasting effects highlight the delicate balance of ABA regulation under HS, where optimal levels enhance tolerance, but excessive or prolonged accumulation compromises rice growth, reproductive development, and yield [7,57,58,59].

Moreover, HS significantly reduces auxin concentrations in rice spikelets and stigmas, which subsequently impedes pollen tube growth and increases the spikelet sterility rate [60,61]. This decline in endogenous auxin levels has been particularly evident in heat-sensitive cultivars, where restricted elongation of pollen tubes within the style is closely associated with auxin deficiency [62]. Exogenous application of synthetic auxins such as naphthaleneacetic acid (NAA) has been shown to partially restore pollen tube growth and reduce sterility, further underscoring the pivotal role of auxin homeostasis in reproductive resilience under thermal stress [63]. Mechanistically, high temperatures are believed to disrupt auxin biosynthesis and polar transport, potentially through altered expression of key auxin biosynthetic genes such as *YUCCA* and *TAA1*, and downregulation of auxin transporters including PIN-FORMED (PIN) proteins [63,64]. Notably, the complex interplay among auxin signaling, ROS accumulation, and peroxidase (POD) activity under high-temperature conditions appears to further compromise pollen tube integrity and directional growth, as ROS-induced oxidative stress can degrade cellular structures and interfere with hormone signaling cascades [65]. Collectively, these findings suggest that the heat-induced disruption of auxin metabolism and signaling is a central regulatory mechanism underlying spikelet sterility in rice, and highlight the potential of auxin-based interventions in enhancing heat tolerance of reproductive tissues [66,67,68].

During the early reproductive stage of rice, HS significantly reduces CTK levels in the panicle, thereby disrupting panicle differentiation and spikelet formation [69]. The homeostasis of CTK is maintained through coordinated regulation of its biosynthesis, transport, and degradation. CTKs are primarily synthesized in the roots and transported to aerial tissues via the xylem, where they play essential roles in panicle development [70]. However, high temperatures impair xylem function, leading to reduced CTK transport efficiency to the panicle. Concurrently, HS induces the activity of cytokinin oxidase/dehydrogenase (CKX), accelerating CTK degradation. Moreover, key enzymes involved in CTK biosynthesis, such as IPT, LOG, and CYP735A, exhibit suppressed activity under HS, further diminishing CTK production and exacerbating hormonal imbalance. This disruption in CTK homeostasis negatively affects branch and spikelet differentiation, panicle exsertion, pollen viability, and anther dehiscence, ultimately leading to reduced spikelet number, smaller panicle size, altered grain morphology, and yield loss [71]. Therefore, sustaining a stable CTK supply and metabolism in the panicle is critical for alleviating the detrimental impacts of HS on rice reproductive development [69,72].

In addition to the major phytohormones discussed above, heat also perturbs the homeostasis of other plant hormones such as gibberellins (GAs), salicylic acid (SA), and brassinosteroids (BRs), further complicating rice’s stress response [73,74,75,76]. These hormone imbalances, through their intricate crosstalk, collectively contribute to impaired growth, reduced fertility, and yield penalties in rice under HS conditions.

## 4. The Cloning of Heat-Tolerance-Related Genes in Rice

The cloning of heat-tolerance-related genes in rice is a key approach to understanding crop adaptation to abiotic stress. This process integrates forward genetics, which uses mapping populations, QTL (quantitative trait loci) mapping, and GWAS to identify genes from natural variation or mutagenesis [77,78], and reverse genetics, which screens candidates from transcriptomic or proteomic data and verifies their functions using gene editing tools such as CRISPR/Cas9. Comparative genomics further accelerates the discovery of conserved heat-responsive pathways through cross-species ortholog analysis. These strategies are supported by techniques including map-based cloning, promoter interaction analysis, and epigenetic profiling. Research has gradually shifted from focusing on individual gene functions to constructing complex regulatory networks. Cloned heat-tolerance genes now serve as valuable targets for marker-assisted breeding and provide a foundation for synthetic biology approaches to enhance crop resilience.

### 4.1. Identification of Heat Tolerance-Related QTLs in Rice

QTL (quantitative trait loci) mapping has been instrumental in elucidating the genetic basis of HS tolerance in rice, with numerous QTLs identified across all 12 chromosomes, influencing both vegetative and reproductive stages (Figure 3). Previous studies have highlighted key QTLs derived from wild rice and heat-tolerant cultivars, such as *qHTH5* (304.2 kb, chromosome 5), *qHTH10* (277.1 kb, chromosome 10), and *qHTB1-1* (47.1 kb, chromosome 1), mapped through linkage analysis in populations from common wild rice [79,80,81]. Heat-tolerant cultivars like IR64 and N22 have been instrumental, revealing major QTLs such as *qHTSF1.1* and *qHTSF4.1*, alongside others like *rlht5.1*, which affects root length under HS. Advanced populations, including chromosome segment substitution lines (CSSLs), have refined QTL mapping, identifying novel loci like *qPSLht4.1* [82,83,84]. Other heat-tolerant varieties like Gan-Xiang-Nuo, Liaoyan 241 and M9962 have also been used to map heat tolerance QTLs. For example, four QTLs for heat tolerance at the flowering stage were mapped between Gan-Xiang-Nuo and the heat-sensitive variety Hua Jing Xian 74 [85]. In a cross between Liaoyan 241 and IAPAR-9, eleven heat-tolerance QTLs were identified, with four major QTLs (*qNS1*, *qNS4*, *qNS6*, *qRRS1*) showing stable detection across environments and interactions with other genes [86,87]. Using QTL-seq, QTLs for spikelet fertility (*qSF1*, *qSF2*, *qSF3*) were found on chromosomes 1, 2, and 3 in the F_2_ population of M9962 and Sinlek [87].

This diagram shows the identified QTLs for heat tolerance in rice, located based on various traits under HS. These traits include seedling survival and growth, flowering time, spikelet fertility, and mature plant traits like grain yield, grain-filling rate, and the frequency of chalky grains (e.g., white-belly, white-base, and milky-white grains).

Further investigations have progressively refined the genetic understanding of HS tolerance across various developmental stages. At the seedling stage, Fan et al. identified three QTLs (*qTT4*, *qTT5*, and *qTT6*) using backcross inbred lines (BILs) between *Oryza longistaminata* and 9311, with *qTT6* corresponding to *OsHSP74.8*, a heat shock protein gene known to stabilize cellular functions under thermal stress [88]. Li et al. conducted a GWAS using ~2.8 million SNPs to identify *qHT7* for seedling-stage heat tolerance, with *OsVQ30* as a proposed candidate gene [89]. At the flowering stage, Stephen et al. employed bulked segregant analysis (BSA) combined with SSR markers and identified RM337, linked to spikelet fertility under HS, with *OsNAC31* emerging as a potential candidate gene [90]. Chen et al. mapped *qHTT8* on chromosome 8 through BSA-seq, identifying *LOC_Os08g07010* and *LOC_Os08g07440* as promising candidates based on differential expression analysis [83]. Meanwhile, specific traits like anther dehiscence under HS have also been dissected: Zhao et al. mapped *qBDL2-2* and *qBDL10*, which control basal dehiscence length, using CSSLs [91]. Hirabayashi et al. developed near-isogenic lines (NILs) incorporating segments from *Oryza officinalis* and mapped *qEMF3*, suggesting that early morning flowering is an adaptive strategy for mitigating HS during anthesis [92]. Hu et al. identified *qRSF9.2*, associated with spikelet fertility, through QTL mapping and GWAS and proposed *LOC_Os09g38500* as a candidate gene with favorable haplotypes [93]. In parallel, grain quality traits under HS have been important targets for genetic studies. Murata et al. mapped *Apq1*, a major QTL that improves grain appearance quality under high temperatures [94], while Miyahara et al. identified two antagonistic QTLs, *qWB6* and *qWB8*, associated with the occurrence of white-back grains [95]. Together, these findings, integrating QTL mapping, GWAS, bulked segregant analysis (BSA), and candidate gene identification, elucidate the genetic basis of HS tolerance across seedling, flowering, and grain quality traits, enabling the development of climate-resilient rice cultivars through marker-assisted selection.

### 4.2. Molecular Cloning of Functional Genes Involved in Heat Tolerance in Rice

Molecular cloning has played a pivotal role in identifying and functionally characterizing genes associated with heat tolerance in rice. Through approaches such as map-based cloning, transcriptome-guided gene discovery, and functional genomics, numerous candidate genes have been isolated and validated. These genes are involved in a wide range of biological processes, including transcriptional regulation, membrane stability, ROS scavenging, signal transduction, photosynthesis and chloroplast stability, hormonal regulation, among others. In combination, these findings have laid a solid foundation for understanding the molecular basis of heat tolerance in rice.

Building on these insights, Table 1 provides a comprehensive summary of key heat-responsive genes identified through molecular cloning, organized by their biological functions and regulatory roles. These genes encompass diverse categories, such as transcription factors (e.g., MYB, NAC), heat-inducible proteins, ubiquitin-related enzymes, membrane-associated components, and hormone-related regulators. Most exhibit positive regulatory effects, enhancing heat tolerance via mechanisms like transcriptional activation, ROS scavenging, and hormone signaling. While the majority function as enhancers of stress resilience, a small subset exerts negative regulation, reflecting the need for precise and context-dependent control of the HS response across developmental stages—including seedling, flowering, and grain-filling phases.

**Table 1 plants-14-02573-t001:** Key genes associated with heat tolerance in Oryza sativa.

Gene	Gene ID	Gene Characteristics	Function Period	Functional Pathway	Regulative Effect	References
*ATT2*	Os03g0856700	Gibberellin 20-oxidase gene	Seedling, reproductive stages	GA signaling	+	[76]
*HTH5*	Os05g0150000	Pyridoxal phosphate homeostasis protein	Seedling	Energy metabolism	+	[79]
*OsMYB55*	Os05g0553400	MYB transcription factor	Seedling	Amino acid metabolism	+	[96]
*OsHSP1*	Os04g0107900	Heat-stimulated protein	Seedling	Molecular chaperone	+	[97]
*SNAC3*	Os07g0225300	NAC transcription factor	Seedling	Antioxidant regulation	+	[98]
*OsANN1*	Os02g0753800	Membrane-bound protein	Seedling	Antioxidant defense	+	[99]
*OsHTAS*	Os09g0323100	Ubiquitin ligase	Seedling	Stomatal closure	+	[100]
*OsTOGR1*	Os03g0669000	DEAD-box RNA helicase	Seedling	RNA helicase	+	[101]
*OsRab7*	Os05g0516600	Small GTPase	Seedling	Antioxidant pathways	+	[102]
*OsHIRP1*	Os03g0302200	E3 ubiquitin ligase	Seedling	Protein ubiquitination	+	[103]
*OsCNGC14*	Os03g0758300	Cyclic nucleotide-gated ion channel	Seedling	Calcium influx	+	[104]
*OsCNGC16*	Os05g0502000	Cyclic nucleotide-gated ion channel	Seedling	Calcium influx	+	[104]
*OsNSUN2*	Os09g0471900	m^5^C RNA methyltransferase	Seedling	RNA methylation	+	[105]
*HTS1*	Os04g0376300	β-ketolipoyl carrier protein reductase	Seedling	Fatty acid synthesis	+	[106]
*TT3.1*	Os03g0706900	E3 ubiquitin ligase	Seedling	Protein ubiquitination	+	[28,107]
*OsSGS3*	Os12g0197500	Gene silencing repressor	Seedling	Gene silencing	+	[108]
*OsUGT72F1*	Os05g0215300	UDP-glucosyltransferase	Seedling	Flavonoid metabolism	+	[109]
*GS2*	Os02g0701300	Growth-regulating factor	Seedling	Hydroxymethyl-glutathione synthetase	+	[110]
*OsIAA7*	Os02g0228900	Auxin-responsive Aux/IAA gene family member	Seedling	Auxin signaling	+	[111]
*HTS2*	Os12g0268000	Cytochrome P450 protein	Seedling	Serotonin biosynthesis	+	[112]
*OsCAF1*	Os10g0412100	Carbon catabolite repressor protein	Seedling	mRNA degradation	+	[113]
*OsDUGT1*	Os01g0597800	Glycosyltransferase	Seedling	Flavonoid glycosylation	+	[114]
*OsPP91*	Os06g0698300	Protein phosphatase	Seedling	Phosphatase activity	+	[115]
*OsEIL5*	Os02g0574800	Transcription factor	Seedling, reproductive stages	Gene activation	+	[115]
*OsRGB1*	Os03g0669200	G protein β subunit	Germination, seedling	Signaling pathways	+	[116]
*OsGRP162*	Os12g0632000	Glycine rich-RNA binding protein	Seedling, booting, fertilization	RNA binding	+	[117]
*OsU2AF35a*	Os09g0491756	Splicing auxiliary factor	Seedling, reproductive stages	RNA splicing	+	[118]
*TT1*	Os03g0387100	α2 subunit of the 26S proteasome	Seedling, flowering, filling	Protein degradation	+	[119]
*HSP24.1*	Os02g0758000	Heat-stimulated protein	Seedling, Flowering, filling	Molecular chaperone	+	[119]
*ERECTA*	Os06g0203800	Receptor kinase	Seedling, flowering	Signaling pathways	+	[120]
*SLG1*	Os12g0588900	Cytoplasmic tRNA 2-thiolated protein	Seedling, flowering	tRNA modification	+	[121]
*OsNCED1*	Os02g0704000	9-cis-Epoxy carotenoid dioxygenase	Flowering	ABA biosynthesis	+	[122]
*Sus3*	Os07g0616800	Sucrose synthase	Filling	Sucrose biosynthesis	+	[123]
*OsGRP3*	Os03g0670700	Glycine-rich RNA-binding protein	Seedling, booting, fertilization	RNA binding	+	[117,124]
*OsPRMT6b*	Os04g0677066	Protein arginine methyltransferase	Seed gemination, growth promotion	ABA receptor degradation	+	[125]
*OsNTL3*	Os01g0261200	NAC transcription factor	Seedling	ER folding	+	[126]
*ATT1*	Os01g0883800	Gibberellin 20-oxidase gene	Seedling, reproductive stages	GA signaling	−	[76]
*TT3.2*	Os03g0707200	Chloroplast precursor protein	Seedling	Chloroplast protection	−	[28,107]
*OsARF6*	Os02g0164900	Auxin response factor	Seedling	Auxin signaling	−	[111]
*OsEBF1*	Os06g0605900	F box protein	Seedling, reproductive stages	Protein degradation	−	[115]
*SCE1*	Os10g0536000	SUMO-conjugating enzyme E2	Seedling, flowering, filling	Protein SUMOylation	−	[119]
*OsFBN1*	Os09g0133600	Ciliary protein	Seedling, flowering, fertilization	Lipid remodeling	−	[127]
*OsMDHAR4*	Os02g0707100	Monodehydroascorbate reductase	Seedling	ROS scavenging	−	[128]
*OsUBP21*	Os11g0573000	Ubiquitin-specific protease	Seedling	Ubiquitin removal	−	[129]
*NAT1*	Os07g0590100	C2H2 family transcription factor	Seedling, reproductive stages	Wax deposition	−	[130]
*HTT1*	Os08g0200100	Stearoyl-acyl carrier protein	Seedling	Lipid saturation	−	[131]
*OsRbohB*	Os09g0438000	NADPH oxidase	Seedling	ROS accumulation	−	[132]
*TT2*	Os03g0407400	G protein γ subunit	Seedling, flowering	Wax biosynthesis	−	[133]
*QT12*	Os12g0173366	Protein transport protein Sec61 subunit beta	Grain quality and yield	ER stress	−	[134]
*NF-YA8*	Os10g0397900	Nuclear transcription factor	Grain quality and yield	Gene transcription	−	[134]

+: positiveregulation; −: negativeregulation.

## 5. Molecular Regulatory Networks of Heat Stress Tolerance in Rice

To unravel the complex molecular mechanisms driving rice HS responses, this section examines the regulatory networks that underpin heat tolerance, highlighting key genes and pathways. As illustrated in Figure 4, these networks orchestrate intricate interactions across plasma membrane signaling, ROS regulation, chloroplast stability, protein homeostasis, transcriptional regulation, and RNA metabolism. The following subsections explore how these molecular components work in concert to bolster rice resilience under HS.

Multiple molecular regulatory pathways contribute to the rice HS response, including ROS homeostasis, chloroplast stabilization, lipid and wax biosynthesis, RNA metabolism, protein degradation, and transcription factor-mediated gene regulation. Calcium-mediated signaling perceives and transmits heat cues, while fatty acid and wax biosynthesis aid membrane remodeling and water retention. Stabilizing chloroplasts preserves photosynthesis, and RNA processing (e.g., mRNA splicing, tRNA modification) maintains transcript stability. The ubiquitin–proteasome system controls protein turnover, and antioxidant pathways sustain ROS balance and mitochondrial function. Transcription factors integrate these signals to coordinate a synchronized defense against HS. Solid arrows indicate positive regulation, dashed arrows indicate potential regulation, and T-shaped arrows indicate negative regulation.

### 5.1. Heat Stress Sensing and Plasma Membrane Integrity

In rice, the PM functions as a frontline sensor of HS, where elevated temperatures disrupt its fluidity and stability, activating membrane-localized calcium-permeable channels that initiate downstream signaling cascades. Among these, OsCNGC14 (cyclic nucleotide-gated ion channel 14) and OsCNGC16, two PM-localized cyclic nucleotide-gated channels, act as critical mediators of calcium influx during thermal stress, translating physical membrane perturbation into intracellular calcium signals. Maintaining PM integrity is essential for effective HS signaling and survival [104]. HTS1 (high temperature sensitive 1), a key enzyme involved in de novo fatty acid biosynthesis, contributes to membrane lipid saturation and fluidity. Loss of HTS1 leads to defective fatty acid production, compromising membrane stability and impairing heat-induced calcium signaling [106]. HTT1 (high temperature tolerance 1), a stearoyl-acyl carrier protein desaturase, reduces PM lipid saturation by catalyzing the conversion of C16:0 to C16:1 fatty acids, thereby negatively affecting heat tolerance. Loss of *HTT1* function enhances heat tolerance, as evidenced by increased PM stability and reduced electrolyte leakage under HS (45 °C) [131]. In parallel, the ERECTA (ER) receptor-like kinase enhances heat tolerance by participating in stress-responsive signaling pathways that help preserve cellular structure and function under HS, although its detailed mechanism in rice remains to be clarified [120]. OsRbohB (respiratory burst oxidase homologue B), a NADPH oxidase located at the PM, is a major source of heat-induced ROS. Under thermal stress, excessive ROS accumulation due to OsRbohB activity can cause membrane lipid peroxidation and structural damage. Reduced expression of *OsRbohB* leads to lower ROS accumulation and mitigates membrane damage under HS, suggesting its role as a negative regulator of heat tolerance [132]. Together, these components constitute a coordinated network in which the PM acts both as a sensor and a target of HS, with lipid metabolism, calcium signaling, and ROS homeostasis jointly determining the membrane’s structural resilience and the plant’s overall heat response.

### 5.2. ROS Accumulation and Antioxidant Regulation

Prolonged HS leads to ROS accumulation in rice, disrupting redox homeostasis and causing oxidative damage. Elevated ROS levels impair membrane integrity, promote lipid peroxidation and protein oxidation, and may trigger PCD [135,136]. To counteract this, rice activates antioxidant defense mechanisms, including enzymatic systems such as superoxide dismutase (SOD) and catalase (CAT), as well as a suite of stress-responsive genes that regulate ROS scavenging and signaling pathways. Several key genes have been identified to mitigate heat-induced oxidative damage by enhancing antioxidant capacity [137,138]. *OsANN1* (*annexin 1*), a calcium-binding annexin, upregulates SOD and CAT expression and alleviates heat-induced membrane injury by boosting ROS scavenging [99]. *MSD1* (*manganese SOD 1*), encoding a manganese superoxide dismutase, catalyzes the conversion of O_2_^−^ to hydrogen peroxide (H_2_O_2_) and activates downstream protective systems including molecular chaperones [45]. The transcription factor *SNAC3* (*stress-responsive NAC1*) plays a critical role in heat-induced ROS regulation by directly activating multiple antioxidant genes, including *CATA*, *RbohF*, and *APX8*, thereby enhancing ROS scavenging capacity and improving HS resistance [98]. Similarly, OsRab7 (Ras-related in brain 7), a small GTPase, promotes antioxidant and stress-responsive pathways, reducing oxidative damage and enhancing survival and yield under HS [102].

Conversely, mutations in *HTS1* or *HTS2* disrupt ROS signaling and detoxification. *HTS1*-deficient plants exhibit increased H_2_O_2_ accumulation and severe cell death, while *HTS2* mutants show impaired serotonin biosynthesis, diminished antioxidant capacity, and ROS accumulation—phenotypes reversible by exogenous serotonin [106,112]. Beyond detoxification, ROS also act as signaling molecules in stomatal and hormonal regulation. The E3 ubiquitin ligase OsHTAS (HEAT TOLERANCE AT SEEDLING STAGE) enhances H_2_O_2_-induced stomatal closure and abscisic acid (ABA) biosynthesis, aiding water retention and HS resistance [100]. In contrast, *OsMDHAR4* (*monodehydroascorbate reductase*), encoding monodehydroascorbate reductase, limits H_2_O_2_ accumulation and suppresses stomatal closure; its downregulation improves heat tolerance by reducing water loss via stomatal regulation [128].

Non-enzymatic antioxidant systems also contribute to ROS homeostasis. The glycosyltransferase OsDUGT1 (Diterpene UDP-glycosyltransferase 1) maintains redox balance by glycosylating flavonoids, which serve as ROS scavengers. Overexpression of *OsDUGT1* enhances heat tolerance by increasing flavonoid-mediated antioxidant capacity, while knockout mutants exhibit higher MDA content, reduced survival, and impaired flavonoid accumulation [114]. Mitochondria, as both sources and targets of ROS, play a crucial role. The mitochondrial protein *HTH5*, encoded by PLPHP, helps stabilize energy metabolism and mitochondrial redox status under HS, thus improving heat tolerance [79]. In addition to classical enzymatic and non-enzymatic systems, UDP-dependent glycosyltransferases (UGTs) have been implicated in heat tolerance by modulating flavonoid metabolism and ROS scavenging. *OsUGT72F1* and *UGT706F1* are strongly induced by HS and contribute to redox homeostasis by enhancing antioxidant enzyme activity and promoting flavonoid glycoside accumulation. Their overexpression improves heat tolerance, while loss-of-function mutants display heightened ROS levels and stress sensitivity. Upstream regulation involves heat shock transcription factors *OsHSFA3* and *OsHSFA4a* for *OsUGT72F1*, and the R2R3-MYB transcription factor *MYB61* for *UGT706F1*, forming transcription-metabolism modules that support ROS scavenging and cellular protection under HS [109,139].

Recent studies also highlight the role of GA signaling in modulating ROS homeostasis. The *ATT1* (*ALKALI-THERMAL TOLERANCE 1*) gene disrupts heat tolerance by inducing large fluctuations in endogenous GA levels, leading to excessive ROS accumulation and oxidative stress. In contrast, *ATT2* enables fine-tuning of GA concentrations to optimal levels, minimizing ROS overproduction while avoiding the negative effects of low GA. Mechanistically, low GA stabilizes DELLA proteins and promotes NGR5 (NITROGEN-MEDIATED TILLER GROWTH RESPONSE 5) accumulation, which enhances H3K27me3 methylation of stress-responsive genes. Thus, ATT2 provides a more effective regulatory mechanism for coordinating hormonal signaling, epigenetic repression, and ROS balance, ultimately improving both heat tolerance and yield in Green Revolution rice varieties [76,140]. These findings underscore a complex and multilayered network involving enzymatic detoxification, transcriptional regulation, secondary metabolism, and hormonal signaling that together govern ROS dynamics and rice resilience under HS.

### 5.3. Photosynthetic and Chloroplast Regulatory Mechanisms

Chloroplast stability is essential for maintaining photosynthetic function under HS. Recent studies have identified a key PM-to-chloroplast signaling module that mediates rice heat tolerance through the *Thermo-tolerance 3* (*TT3*) locus. This locus comprises TT3.1, a PM-localized E3 ubiquitin ligase, and TT3.2, a chloroplast precursor protein. Under HS, TT3.1 moves to endosomes and ubiquitinates TT3.2, leading to its vacuolar degradation. This process limits the accumulation of mature TT3.2 in chloroplasts and is critical for protecting thylakoids from thermal damage. The TT3.1-TT3.2 module thus serves as a key signaling pathway that transduces external heat cues to the chloroplast, safeguarding photosynthetic machinery and reducing yield loss under high-temperature conditions [28,107]. In parallel, RNA methylation also plays a role in maintaining chloroplast function during HS. OsNSUN2 (NOP2/Sun RNA methyltransferase), a 5-methylcytosine (m^5^C) RNA methyltransferase, facilitates mRNA modifications that enhance the translation of proteins involved in photosynthesis and detoxification, helping sustain chloroplast activity and redox balance [105]. Additionally, antioxidant regulation is critical for chloroplast protection. OsMDHAR4, a chloroplast-localized monodehydroascorbate reductase, is highly expressed in leaves, where it contributes to ROS scavenging. Loss of *OsMDHAR4* reduces water loss and improves heat tolerance, highlighting its role in redox homeostasis. Lipid metabolism also directly affects chloroplast membrane stability. *HTS1*, encoding a ketoacyl carrier protein reductase (KAR), is localized to the thylakoid membrane and supports membrane integrity. The *hts1* mutant exhibits severe chloroplast damage and accelerated senescence under HS. Moreover, OsFBN1 (fibrillin 1), a plastid-lipid-associated fibrillin protein, regulates chloroplast lipid remodeling by binding C18- and C20-fatty acids and promoting plastoglobule (PG) formation. *OsFBN1* overexpression alters the expression of genes related to jasmonic acid biosynthesis, thylakoid stability, photosynthesis, and isoprenoid metabolism but also impairs grain filling and panicle development under HS [127]. Collectively, these findings highlight a complex regulatory network involving heat signal transduction, post-transcriptional modification, antioxidant defenses, and lipid metabolism that coordinately preserves chloroplast stability and photosynthetic efficiency in rice under high-temperature conditions.

### 5.4. Protein Homeostasis Regulation

Heat stress severely disrupts protein homeostasis, threatening both cell viability and crop productivity. Under high temperatures, misfolded and denatured proteins accumulate rapidly, requiring efficient clearance to prevent cytotoxicity. In rice, the ubiquitin/26S proteasome system plays a central role in maintaining proteostasis. A key heat tolerance QTL, *TT1* (*Thermo-tolerance 1*), encodes the α2 subunit of the 26S proteasome, promoting the degradation of heat-damaged proteins. Comparative studies have shown that *OgTT1* (*Oryza glaberrima TT1*) is more effective than its *Oryza sativa* homolog (*OsTT1*) in mitigating protein aggregation and sustaining stress responses. Overexpressing *OgTT1* enhances heat tolerance across multiple species, including rice, Arabidopsis, and *Festuca elata*, providing a valuable resource for breeding climate-resilient crops. Functionally, *SCE1* (*small ubiquitin-like modifier-conjugating enzyme 1*), acting downstream of *TT1*, negatively regulates heat tolerance by mediating the SUMOylation of small heat-shock proteins (sHSPs) such as Hsp24.1 (heat shock protein 24.1) and Hsp40 (heat shock protein 40). Excessive SUMOylation of sHSPs leads to their abnormal accumulation, impairing heat tolerance. Loss of *SCE1* alleviates this effect, promoting higher grain filling and seed-setting rates under HS [119,141]. In parallel, OsHIRP1 (heat-induced RING finger protein 1) and OsHTAS (heat tolerance-associated RING finger E3 ligase), both RING-type E3 ligases, facilitate ubiquitination of denatured proteins, while OsUBP21 (Ubiquitin-specific protease 21) counteracts this process by removing ubiquitin chains, highlighting the importance of balanced ubiquitination and deubiquitination in stress adaptation [103,129]. Meanwhile, hormonal signaling pathways integrate with proteostasis to regulate recovery after stress. *OsPRMT6b* (*protein arginine methyltransferase 6b*) promotes the degradation of the ABA receptor OsPYL/R10 (PYRABACTIN RESISTANCE-LIKE/REGULATORY COMPONENT OF ABA RECEPTOR, R10) via methylation, facilitating its ubiquitination by Tiller Enhancer (TE) and attenuating ABA signaling to enable post-stress growth. In *osprmt6b* mutants, delayed ABA receptor degradation prolongs stress responses and hampers growth recovery [125]. Similarly, the *OsEBF1–OsEIL5–OsPP91* module fine-tunes heat tolerance by coordinating protein turnover and gene expression: *OsEBF1* (*EIN3-binding F-box protein 1*) mediates the ubiquitin-dependent degradation of OsEIL5 (ETHYLENE INSENSITIVE3-LIKE 5), a transcription factor that activates *OsPP91* (*protein phosphatase 91*), a type 2C phosphatase essential for heat tolerance. Together, these mechanisms reveal a complex regulatory network integrating ubiquitin-proteasome-mediated degradation, SUMOylation, translational control, and hormonal signaling, providing new avenues for improving crop resilience.

### 5.5. Regulatory Network of Transcription Factors

High temperature stress severely disrupts rice growth and productivity, triggering a complex transcriptional regulatory network to maintain cellular stability and improve heat tolerance. Among these mechanisms, wax metabolism plays a crucial protective role. The *TT2* (*heat tolerance 2*)-*SCT1* (*Sensing Ca^2+^ transcription factor 1*)-*OsWR2* (*wax synthesis regulator 2*) module links G protein signaling, calcium decoding, and wax biosynthesis. Under HS, TT2 disruption reduces calcium influx, which weakens the calmodulin–SCT1 interaction. This, in turn, lifts SCT1-mediated repression of *OsWR2*, promoting wax accumulation and enhancing heat tolerance [133]. Similarly, the C2H2-type transcription factor *NAT1* (*negative regulator of heat tolerance 1*) negatively regulates *bHLH110* (*basic helix-loop-helix 110*), a direct activator of wax biosynthetic genes *CER1* (*ECERIFERUM1*) and *CER1L* (*CER1-like*). Loss of *NAT1* removes this repression, enhancing wax deposition and heat tolerance in the field. In parallel, rice coordinates additional transcriptional responses to counteract HS [130]. The membrane-associated NAC transcription factor *OsNTL3* (*NAC Transcription factor-Like 3*) translocates from the PM to the nucleus upon HS, where it activates ER protein folding genes and directly regulates *OsbZIP74* (*basic leucine zipper 74*), forming a feedback loop that integrates ER, membrane, and nuclear signaling [126]. Auxin signaling also contributes to heat tolerance through the *OsIAA7* (*indole-3-acetic acidiInducible 7*)-*OsARF6* (*auxin response factor 6*)-*OsTT1*/*OsTT3.1* pathway, where *OsIAA7* inhibits *OsARF6*, derepressing *OsTT1* and *OsTT3.1* expression to promote heat resistance [111]. A heat-responsive regulatory module involving NF-Y transcription factors and *QT12* underlies the difference in heat tolerance between *O. sativa* var. *indica* and japonica rice. In heat-sensitive *O. sativa* var. *indica* and *japonica* varieties, NF-YA8 binds to the CCAAT-box in the *QT12* promoter and increases its expression. In contrast, NF-YB9 and NF-YC10 inhibit this binding, helping to preserve ER homeostasis and maintain a balance between storage protein and starch synthesis. Under HS, NF-YA8–NF-YB9/YC10 interactions weaken, leading to *QT12* activation, which suppresses *IRE1*, triggers ER stress and UPR, reduces protein accumulation, and increases chalkiness. In thermotolerant *O. sativa* var. *indica*, a G/A variation disrupts the CCAAT-box, preventing NF-YA8 binding and keeping *QT12* expression low, thereby maintaining ER stability and heat tolerance. This mechanism, defined as heat tolerance-related haplotypes (TRHs), provides a genetic basis for *indica–japonica* differentiation in heat response during grain filling [134,142]. The *MYB61-UGT706F1* module, involving *MYB61* (*MYB transcription factor 61*) and *UGT706F1* (*UDP-glucosyltransferase 706F1*), further promotes ROS scavenging by enhancing flavonoid glycosylation and antioxidant capacity [139]. Moreover, *OsUGT72F1* (*UDP-glucosyltransferase 72F1*), transcriptionally regulated by *OsHSFA3* (*heat shock factor A3*) and *OsHSFA4a* (*heat shock factor A4a*), further contributes to ROS homeostasis by modulating phenylpropanoid, zeatin, and flavonoid metabolism [109]. In addition, *OsMYB55* (*MYB transcription factor 55*) activates amino acid metabolic genes including *OsGS1;2* (*glutamine synthetase 1;2*), *GAT1* (*glutamine amidotransferase 1*), and *GAD3* (*slutamate decarboxylase 3*), leading to the accumulation of stress-related amino acids such as glutamate, GABA, and arginine, which further enhance heat tolerance [96]. In combination, these findings reveal a complex transcriptional network underlying rice HS responses. This network integrates multiple regulatory pathways, such as wax metabolism, ER stress regulation, hormone signaling, ROS homeostasis, and amino acid biosynthesis, to establish a coordinated and robust defense system against high-temperature stress.

### 5.6. RNA Metabolism Regulation and Response Mechanisms

Heat stress severely impacts rice production, making the understanding of its genetic basis crucial for improving heat tolerance. In this study, several key components of RNA metabolism were found to regulate HS responses in rice. The gene *OsU2AF35a* (*U2 snRNP auxiliary factor 35a*), a critical part of the splicing complex, is induced by HS and plays an essential role in heat tolerance by promoting proper RNA splicing and phase separation of proteins under heat conditions. Mutants lacking *OsU2AF35a* exhibited reduced oxidative stress resistance and impaired splicing of key heat-responsive genes, highlighting its importance in regulating the stress response [118]. Additionally, the DEAD-box RNA helicase *OsTOGR1* (*thermotolerant growth required 1*) was shown to improve HS tolerance in transgenic non-heading Chinese cabbage [101]. Similarly, the glycine-rich RNA-binding protein *OsGRP3* (*glycine-rich RNA-binding protein 3*) enhanced drought resistance in rice by modulating the phenylpropanoid biosynthesis pathway and increasing lignin accumulation [124]. Furthermore, a trade-off between heat tolerance and disease resistance was revealed through the interaction of *OsSGS3a* (*Suppressor of gene silencing 3a*) with its homolog *OsSGS3b* (*Suppressor of gene silencing 3b*), which regulates the biogenesis of trans-acting small interfering RNAs (tasiRNAs) targeting auxin response factors (ARFs). This balance between abiotic and biotic stress responses underscores the complexity of plant stress tolerance mechanisms [108]. The tRNA 2-thiolation protein *SLG1* (*Slender Guy 1*) was found to play a critical role in heat tolerance, particularly in *indica* rice varieties, by increasing heat tolerance through enhanced tRNA modification [121]. The CCR4-NOT complex subunit *OsCAF1A* (*CCR4-associated factor 1A*) also contributes to HS tolerance by regulating mRNA degradation and turnover, with its relocalization to processing bodies being regulated by DEAD-box RNA helicases [113]. These findings collectively underscore the pivotal role of RNA metabolism in regulating plant responses to HS.

## 6. Conclusions and Future Directions

High temperature stress remains one of the most critical environmental constraints on rice production worldwide, exacerbated by the increasing frequency and intensity of extreme heat events under climate change. Its impact extends across multiple growth stages, threatening both yield and grain quality, and posing serious challenges to hybrid seed production. While significant progress has been made in elucidating the physiological, molecular, and genetic basis of thermotolerance, the effective translation of these findings into breeding solutions that are robust across diverse environments remains a formidable task.

### 6.1. Breeding Strategies Utilizing Existing Resources

Accelerating the development of thermotolerant rice cultivars requires the strategic use of existing genetic resources. Natural germplasm, including wild rice species and landraces, offers abundant allelic variation for heat adaptation. Several functional genes associated with heat tolerance, such as *TT1*, *QT12*, *NAT1*, and *TT3.1* [28,119,130,134], have been identified and represent valuable targets for improvement.

A practical approach is to combine multiple favorable alleles into elite cultivars through gene pyramiding. This can be achieved by crossing high-performing but heat-sensitive cultivars with thermotolerant donor lines, followed by marker-assisted selection (MAS) or genomic selection (GS) to track and select desirable alleles. High-throughput phenotyping platforms further facilitate large-scale, non-destructive evaluation of traits such as canopy temperature, chlorophyll fluorescence, and spikelet fertility under both field and controlled-environment conditions. Emerging approaches, including speed breeding, artificial intelligence (AI)-driven parental selection, and gene editing, offer additional avenues for enhancing thermotolerance in elite cultivars while shortening breeding cycles. Despite these promising strategies, their effectiveness can be constrained by variable performance of heat-tolerant genes across environments, the underutilization of wild germplasm due to linkage drag and poor adaptation, and the lengthy breeding cycles required for introgression. Addressing these limitations requires a more integrated view of the technical, biological, and environmental barriers that influence the success of functional genomics in practical breeding.

### 6.2. Translational Challenges in Breeding Heat-Tolerant Rice

While advances in molecular marker technology, genomic selection, and precision gene editing provide powerful opportunities to accelerate the development of heat-tolerant rice cultivars, their practical translation into breeding programs remains challenging. Heat tolerance is inherently polygenic, and most favorable alleles exert only small individual effects, making it difficult to achieve substantial gains through single-gene interventions. The utilization of valuable germplasm is further complicated by adaptation gaps between donor and recipient genotypes, which can compromise hybrid performance, and by background genetic noise that may mask the effects of beneficial alleles. Findings from laboratory studies often fail to perform consistently in the field because they lack large-scale testing across multiple environments. The transition from gene discovery to stable cultivar release is also hindered by limitations in transformation efficiency, trait stability, and genotype compatibility. Beyond these technical constraints, external factors such as regulatory frameworks, societal acceptance of biotechnology-derived varieties, and restricted access to advanced phenotyping platforms will strongly influence the pace and extent of adoption.

### 6.3. Future Research Directions

Looking forward, breeding programs must expand the genetic toolbox by identifying novel genes and regulatory loci associated with heat tolerance. Establishing stable high-temperature screening nurseries in controlled-environment greenhouses is strongly recommended to precisely simulate field HS conditions. These platforms will enable systematic evaluation of diverse germplasm, with selected materials serving as essential resources for genomic mapping, haplotype mining, and molecular marker development, thereby facilitating the discovery of new heat-tolerance genes and elite haplotypes.

Research should also target less-understood regulatory layers that fine-tune plant responses to HS. These include epigenetic modifications such as DNA methylation, histone modifications, chromatin remodeling, and regulation by non-coding RNAs. Epigenetic regulation can rapidly and reversibly reprogram gene expression without altering the DNA sequence, influencing thermotolerance and potentially establishing stress memory. Signal transduction pathways, including calcium-mediated signaling, mitogen-activated protein kinase (MAPK) cascades, and heat shock factor (HSF) networks, remain incompletely characterized in rice and require deeper functional dissection to understand how they integrate environmental cues into transcriptional and metabolic responses. Hormonal crosstalk—involving abscisic acid (ABA), auxin, cytokinins (CKs), gibberellins (GAs), and brassinosteroids (BRs)—also plays a central role in coordinating growth–defense trade-offs under HS and deserves further mechanistic exploration. Additionally, the interplay among ROS homeostasis, redox signaling, and antioxidant defenses warrants comprehensive investigation, as these pathways are tightly linked to both signal perception and downstream protective responses. Integrating epigenome-wide profiling, transcriptomics, phosphoproteomics, and metabolomics will enable the identification of key regulatory hubs and their dynamic interactions. Such multi-layered insights, combined with advanced genome editing and epigenome editing technologies, could provide new opportunities for targeted breeding of cultivars with durable heat resilience.

In parallel, genome editing technologies such as CRISPR/Cas systems present unprecedented opportunities for targeted improvement. Precise editing of regulatory elements and candidate genes can create tailored alleles with enhanced stress resilience while minimizing linkage drag and unintended effects. Integrating genome editing with high-throughput phenotyping and genomic selection can streamline the development of cultivars capable of maintaining yield and quality under extreme heat. Future strategies will likely employ multiplexed genome editing to simultaneously enhance multiple pathways, reflecting the polygenic nature of heat tolerance.

By broadening the scope of regulatory studies and harnessing cutting-edge breeding tools, it will be possible to develop next-generation rice cultivars that are resilient to rising global temperatures while sustaining productivity, grain quality, and hybrid seed production efficiency.

## Figures and Tables

**Figure 1 plants-14-02573-f001:**
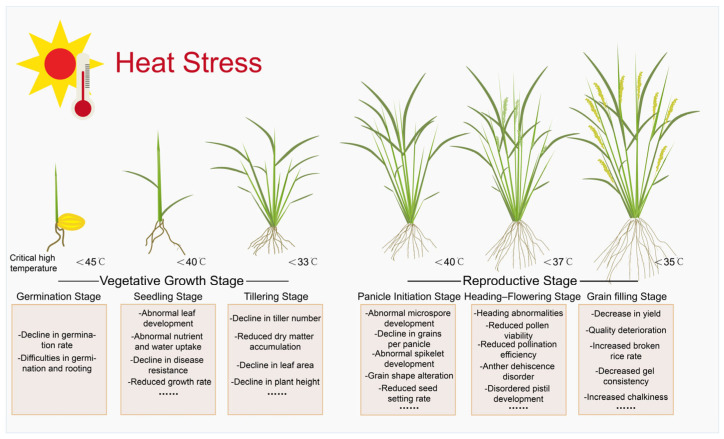
Impact of HS on rice agronomic traits at different stages.

**Figure 2 plants-14-02573-f002:**
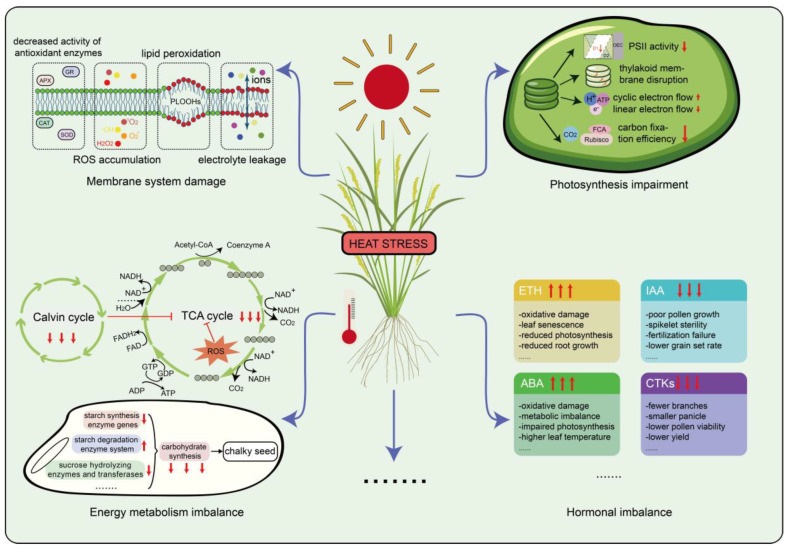
Overview of physiological alterations in rice under HS.

**Figure 3 plants-14-02573-f003:**
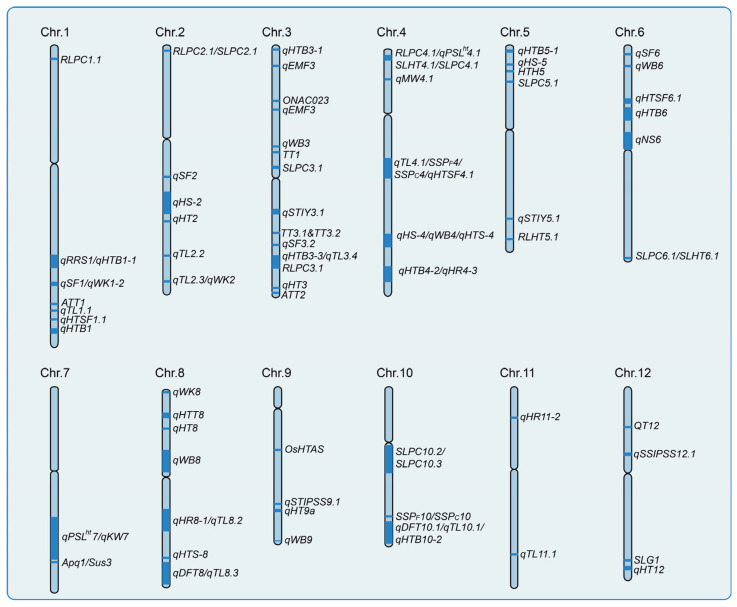
Genetic loci (QTLs) for heat stress response in rice.

**Figure 4 plants-14-02573-f004:**
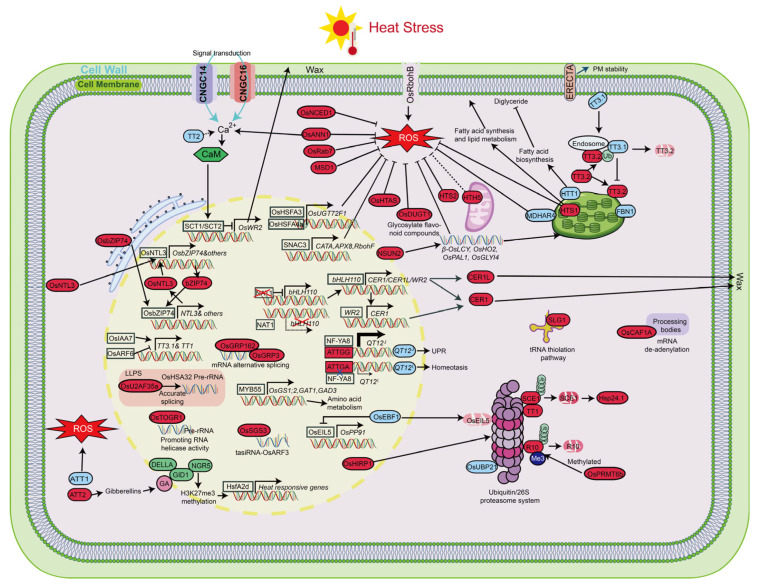
Molecular pathways governing heat stress resistance genes in rice.

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
