# Peer review of "Rice Heat Stress Response: Physiological Changes and Molecular Regulatory Network Research Progress"

_plants, 2025, doi:10.3390/plants14162573_

Round 1

Reviewer 1 Report

Comments and Suggestions for Authors

Firstly, I would like to thank all the authors for their valuable contribution in reviewing the impact of heat stress in rice, a globally significant crop. The manuscript comprehensively covers various aspects of heat stress responses in rice. However, a few improvements are suggested to enhance the manuscript’s appeal and relevance to a broader global audience. Kindly consider the following comments:

Impact of Heat Stress on the Agronomic Performance of Rice (Line 51):

It is suggested that the authors present a tabulated summary of the critical temperature thresholds across various developmental stages of rice, along with the associated physiological, morphological, and yield-related consequences. This addition would enhance the utility of the manuscript as a reference for researchers and agronomists.

The Cloning of Heat-Tolerance-Related Genes in Rice (Line 347):

The manuscript would benefit from a detailed table outlining key quantitative trait loci (QTLs) linked to heat tolerance in rice, including information on associated traits, mapping populations, and chromosomal locations. Furthermore, the inclusion of recent advancements in genome editing technologies, particularly CRISPR-Cas systems, and their applications in enhancing heat tolerance, is highly recommended to reflect the current state of the art.

Conclusions and Future Directions (Line 680):

  1. To strengthen this section, the authors are encouraged to incorporate emerging approaches such as high-throughput phenotyping, speed breeding, and the integration of artificial intelligence (AI) in climate-resilient rice breeding programs. These technologies represent transformative tools in accelerating varietal development under heat stress conditions.

Author Response

Comment 1:
It is suggested that the authors present a tabulated summary of the critical temperature thresholds across various developmental stages of rice, along with the associated physiological, morphological, and yield-related consequences.

Response:

Thank you for your insightful and valuable feedback. We greatly appreciate your suggestion to present a tabulated summary of the critical temperature thresholds and their associated consequences. This is an important point that highlights the need for clarity and a comprehensive overview. We have carefully considered this and agree on the importance of these indicators. The physiological, morphological, and yield-related consequences of high-temperature stress are key aspects of our study, and we have already integrated this information into Figure 1. To avoid redundancy and keep the manuscript focused, we have chosen not to create an additional table.

Instead, in response to your comment, we have revised Figure 1 to more clearly and explicitly label the specific high-temperature critical temperatures for each rice growth stage. We believe this modification improves the clarity of the figure and effectively addresses your suggestion without duplicating information.

Comment 2:
The manuscript would benefit from a detailed table outlining key QTLs linked to heat tolerance in rice, including information on associated traits, mapping populations, and chromosomal locations. Furthermore, include recent advancements in genome editing technologies, particularly CRISPR-Cas systems.

Response:
Thank you for your insightful suggestion to include a detailed table outlining key Quantitative Trait Loci (QTLs) linked to heat tolerance in rice, as well as recent advancements in genome editing technologies, particularly CRISPR-Cas systems. We greatly appreciate your feedback, which enhances the scientific rigor and clarity of our manuscript.

In response to your recommendation, we have included Figure 3 | Genetic Loci (QTLs) for Heat Stress Response in Rice, which presents a comprehensive genome-wide map of QTLs associated with heat tolerance in rice, including their chromosomal locations. Furthermore, we have incorporated a discussion on recent advancements in CRISPR-Cas systems and their applications in improving heat tolerance in rice in lines 687–693.

Comment 3:

Incorporate emerging approaches such as high-throughput phenotyping, speed breeding, and the integration of AI in climate-resilient rice breeding programs.

Response: We have revised the “Future Research Directions” section to highlight emerging breeding technologies, including high-throughput phenotyping, speed breeding, and AI-assisted predictive modeling for climate-resilient rice improvement (lines 634–646).

Reviewer 2 Report

Comments and Suggestions for Authors

This is a comprehensive and informative review focusing on rice responses to heat stress, covering physiological responses, hormonal regulation, functional gene identification, and molecular regulatory networks. It is well-structured and references many relevant studies. However, it still requires improvement in methodological clarity, figure-text integration, and language fluency.

1As a review paper, please clarify the methodology used for literature selection (e.g., databases, time span, inclusion criteria).

2Consider categorizing key genes in Table 1 more systematically by functional pathways to enhance clarity.

3Figure 3 is informative but complex; please provide more structured explanation in the text to match the figure’s content.

4A more explicit discussion on the current limitations in molecular integration, germplasm application, and breeding translation is needed.

5Please consider adding a dedicated subsection on epigenetic regulation under heat stress.

6Please structure the “Future Directions” more clearly into key themes such as gene mining, regulatory network modeling, and breeding strategies.

7Some sentences are overly complex; professional language editing is recommended to improve fluency.

8Include more recent references (2022–2024) to reflect the current state of the field.

9A short section on the applied prospects and challenges of translating functional gene findings into breeding is recommended.

Author Response

Comment 1:

Please clarify the methodology used for literature selection.

Response:

We agree that the methodology should be clearly stated. We have added a new paragraph in the Introduction section to outline our literature search strategy (lines 48–56). This now includes the databases we used (e.g., PubMed, Web of Science), the time span for the search, and the inclusion and exclusion criteria for selecting relevant studies.

Comment 2:

Categorize key genes in Table 1 more systematically by functional pathways.

Response:

Thank you for your valuable suggestion to categorize key genes in Table 1 more systematically by functional pathways. We greatly appreciate your feedback, which helps improve the clarity and quality of our manuscript.

In response to your comment, we have revised Table 1 by adding a new column titled "Functional Pathway" to annotate the molecular function of each gene. This addition provides further clarity on the roles of the genes included in the table.

Comment 3:

Figure 3 is informative but complex; please provide a more structured explanation in the text.

Response:

We appreciate your insight regarding the complexity of Figure 3. To facilitate a comprehensive understanding, the text in the "Molecular Regulatory Networks of Heat Stress Tolerance in Rice" section was designed to provide a structured explanation. The functions of the genes in the figure are clustered and analyzed based on their involvement in several key pathways, such as signal transduction, membrane damage, ROS, chloroplast homeostasis and photosynthesis, protein homeostasis regulation, RNA metabolism, and transcriptional regulation. This detailed organization was implemented to ensure a clear and direct integration of the figure's content with the narrative.

Comment 4:

A more explicit discussion on current limitations in molecular integration, germplasm application, and breeding translation is needed.

Response:

We appreciate you highlighting the need for a more explicit discussion of the current limitations. We agree that this is a crucial point for a forward-looking review. In response, we have added a new subsection to the "Future Directions" section of the manuscript (lines648–660). This part now provides a more detailed discussion of the challenges and bottlenecks in molecular integration, germplasm application, and the practical translation into breeding programs.

Comment 5:

Add a dedicated subsection on epigenetic regulation under heat stress.

Response:

Thank you for your valuable suggestion. We agree that epigenetic regulation is a crucial and emerging area of research. We would like to clarify that in section 5.6 RNA Metabolism Regulation and Response Mechanisms, we have already discussed genes such as OsSGS3a, OsSGS3b, and SLG1, which are known to be involved in the epigenetic regulation process. While other genes in that section (e.g., OsU2AF35a, OsTOGR1, OsGRP3, OsCAF1A) primarily function in RNA metabolism, they may also indirectly influence the epigenetic state. Your reminder about the immense research value of this topic is excellent, and following your suggestion, we have now explicitly highlighted this point as a key area for future study within section 6.3 Future Research Directions.

Comment 6:

Structure the “Future Directions” into key themes.

Response:

Thank you for your valuable suggestion. We agree that a more structured "Future Directions" section greatly enhances clarity for the reader.

In response to your comment, we have revised the "Conclusions and Future Directions" section to be more thematic. It is now divided into three distinct subsections to better organize the content: 6.1 Breeding Strategies Utilizing Existing Resources; 6.2 Translational Challenges in Breeding Heat-Tolerant Rice; 6.3 Future Research Directions. We believe this new structure effectively addresses your suggestion and provides a clearer roadmap for the field.

Comment 7:

Some sentences are overly complex; professional language editing is recommended.

Response:

We appreciate your helpful suggestion. We've carefully reviewed the entire manuscript and revised several overly complex sentences to improve their clarity and readability (e.g., lines44-47; lines222-227; lines497-499; lines572-577).

Comment 8:

Include more recent references (2022–2024).

Response:

Thank you for your valuable suggestion. We have made every effort to include the majority of recent publications related to rice heat stress responses, with 30 references from 2022–2024. If you have specific additional references in mind, we would greatly appreciate your guidance and will gladly incorporate them.

Comment 9:

Add a short section on applied prospects and challenges of translating functional gene findings into breeding.

Response:

Thank you for your valuable suggestion. In response, we have added a new section, 6.2 Translational Challenges in Breeding Heat-Tolerant Rice, which addresses the applied prospects and challenges of translating functional gene findings into breeding programs for heat-tolerant rice cultivars.

Reviewer 3 Report

Comments and Suggestions for Authors

Manuscript: Rice Heat Stress Response: Physiological Changes and Molecular Regulatory Network Research Progress

Written by: WeiWei MA 1, †, XiaoLe WANG 1, †, ChuanWei GU 3, †, ZhengFei LU 1, RongRong Ma 1, 2, XiaoYan WANG 1, 2, YongFa LU 1, 2, KeFeng CAI 1, 2, ZhiMing TANG 1, 2, ZhuoQi ZHOU 2, ZhiXin CHEN 2, HuaCheng ZHOU 1, 2, * and XiuHao BAO 1, *

The review article provides an overview of how heat stress affects the agronomic traits of rice at different stages of development. The authors decribed the main physiological and metabolic changes induced by heat stress in rice. The article is well written and divided into useful sections, but needs to be improved.

I suggest a title: ​“Response of Rice to Heat Stress: Physiological Changes and Molecular Regulatory Networks - Research Progress”. In my opinion, some parts need to be rewritten as there are long parts referring to a single citation (e.g. lines 84-92, 122-135, 135-146…). Please follow the format precribed by the journal for illustrations. The figures should be placed in the main text close to the first citation. I therefore suggest placing them before sections 2.1. (Fig. 1), 3.1. (Fig. 2), 5.1. (Fig. 3). I also suggest to shortening the captions and not repeating what is written below. Please correct the abbreviations throughout the manuscript, as the full name should be given first and then the abbreviation (e.g. in lines 232-242;495; lines 446,449, 456…). For example, PM is already defined in line 164. Also, a sentence should not start with an abbreviation (line 473, 475), endoplasmic reticulum is mentioned in line 736 – the abbreviation used in line 615, etc. Please follow format required by journal for references. For example, in reference no. 1., 9. the title of the source is missing, while in reference 13. doi is missing, etc.; the page number there should be followed by a full stop instead of a comma. Further suggestions can be found below.

After editing the attached comments, the manuscript can be published in Plants.

Suggestions for improving the paper:

Please - Rewrite additionally, in addition (lines 191 and 194), at the end of almost every subsection “together, collectively, and in addition”, are used very often.

Line 22: Keywords – “Rice” with capital R

Line 164 – HS is defined, bud abbreviation should be defined at first mention, which is in the introduction section – line 28, please revise.

Line 439 – Plasma membrane – the abbreviation is already defined in line 164, so use PM instead.

Line 361 – QTL abbreviation is used, but the full name is in line 361, please correct.

4.2. It seems that there are two spaces before each line in this part.

Line 466 – duplication of “heat stress”, duplication ROS overaccumulating in rice – the same in line 255- Figure caption.

Line 618 O. sativa var. indica?

Author Response

Comment 1:

Suggested title change.

Response:

We thank the reviewer for the suggestion. We have revised the title to “Response of Rice to Heat Stress: Physiological Changes and Molecular Regulatory Networks – Research Progress,” which better reflects the manuscript’s scope.

Comment 2:

Avoid long sections referring to a single citation.

Response:

We thank the reviewer for the helpful comment. We have revised the manuscript to reduce over-reliance on single references in long sections, incorporating multiple relevant citations where appropriate (lines135-141, lines141-147).

Comment 3:

Follow the journal’s figure placement rules and shorten captions.

Response:

We are grateful for the reviewer’s insightful suggestion. We have adjusted figure placement to appear close to their first mention (Fig. 1 before Section 2.1, Fig. 2 before Section 3.1, Fig. 3 before Section 5.1) and have shortened captions to avoid repetition of the main text.

Comment 4:

Correct abbreviations and ensure proper first-use definition.

Response:

We have checked all abbreviations to ensure that full terms are provided at first mention and abbreviations are used consistently thereafter. Sentences no longer start with abbreviations.

Comment 5:
Follow journal reference style.

Response:
We thank the reviewer for providing this valuable recommendation. We have reformatted all references to comply with the journal’s requirements, correcting missing titles, DOIs, and punctuation inconsistencies.

Comment 6:

Rewrite additionally, in addition (lines 191 and 194), at the end of almost every subsection together, collectively, and in addition, are used very often.

Response:

We have removed or replaced redundant phrases to improve clarity and variety.

Comment 7:

Suggestions for improving the paper:

Response:

Keyword capitalization: Corrected “rice” to “Rice” in the keywords section (Line 22).

Definition of HS: Moved the definition of “HS” to its first occurrence (Line 25).

Plasma membrane abbreviation: Replaced all “plasma membrane” with “PM”, as it was already defined earlier.

QTL abbreviation order: Corrected the order of “quantitative trait loci (QTL)” in Line 341.

Formatting in Section 4.2: Removed the unintended double spaces at the beginning of each line in Section 4.2.

Repetitive wording: Deleted repeated phrases “heat stress” and “ROS overaccumulating in rice” at Line 466.

Scientific naming: Verified and corrected the writing of O. sativa var. indica at Line 618 according to standard nomenclature rules.

Round 2

Reviewer 2 Report

Comments and Suggestions for Authors

Accept in current form.

Reviewer 3 Report

Comments and Suggestions for Authors

After incorporating my previous suggestions and minor edits, I consider this article suitable for publication in the journal Plants.